# Water Management Capacity of Metal Foam Flow Field for PEMFC under Flooding Situation

**DOI:** 10.3390/mi14061224

**Published:** 2023-06-10

**Authors:** Lingjiang Chen, Zichen Wang, Chuanfu Sun, Hui Zhu, Yuzhen Xia, Guilin Hu, Baizeng Fang

**Affiliations:** 1School of Mechanical and Energy Engineering, Zhejiang University of Science and Technology, Hangzhou 310023, China; 2Department of Energy Storage Science and Technology, University of Science and Technology Beijing, 30 College Road, Beijing 100083, China; baizengfang@163.com

**Keywords:** fuel cell, flow field, porous metal foam, water management, self-humidification

## Abstract

Porous metal foam with complex opening geometry has been used as a flow field to enhance the distribution of reactant gas and the removal of water in polymer electrolyte membrane fuel cells. In this study, the water management capacity of a metal foam flow field is experimentally investigated by polarization curve tests and electrochemical impedance spectroscopy measurements. Additionally, the dynamic behavior of water at the cathode and anode under various flooding situations is examined. It is found that obvious flooding phenomena are observed after water addition both into the anode and cathode, which are alleviated during a constant-potential test at 0.6 V. Greater abilities of anti-flooding and mass transfer and higher current densities are found as the same amount of water is added at the anode. No diffusion loop is depicted in the impedance plots although a 58.3% flow volume is occupied by water. The maximum current density of 1.0 A cm^−2^ and the lowest *R*_ct_ around 17 mΩ cm^2^ are obtained at the optimum state after 40 and 50 min of operation as 2.0 and 2.5 g of water are added, respectively. The porous metal pores store a certain amount of water to humidify the membrane and achieve an internal “self-humidification” function.

## 1. Introduction

Polymer electrolyte membrane fuel cells (PEMFCs) are very promising electrical energy sources for many applications, especially for automotive applications [1]. They have been intensively studied because of their many advantages such as zero-emission, high energy conversion efficiency (theoretical conversion efficiency up to 85%), high power density, and fast start-up at low temperatures [2,3,4,5,6,7]. While polymer membranes such as Nafion^®^ and Dow^®^ are commonly used as an electrolyte in PEMFCs, water in a bound state within the membrane has a positive effect on the cell performance by facilitating proton transfer from the anode to the cathode [8,9]. A low water content results in a high resistance, which has a negative effect on the fuel cell operation [10]. In a flood situation, a high content of water blocks the diffusion path of the reactant and, therefore, a high mass transfer resistance and poor PEMFC performance would be obtained [11]. In order to ensure the fuel cell output performance and service life, it is necessary to maintain the membrane in a highly hydrated state and prevent flooding. However, water transport in fuel cells is a complex and dynamic process, which involves a variety of mechanisms, including adsorption, diffusion, conduction, and migration. Numerous studies have been conducted to understand the water migration mechanisms and describe the water transport process.

An ideal flow field for PEMFCs, with high electrical and thermal conductivity, plays an important role in homogeneous gas distribution and the efficient removal of excess water [12]. The commonly used flow fields, such as parallel, serpentine, interdigitated, and multi-serpentine flow fields, are widely discussed for their water management capabilities. The serpentine flow field is reported to exhibit better mass transfer performance and less water accumulation compared to the parallel flow field [13]. The interdigitated flow field is proven to remove water from the flow channel more efficiently [14]. However, these designs suffer from a non-uniform gas and temperature distribution, low catalyst usage, and difficulty in removing liquid water from the flow channels. To alleviate the situation of water deficit in the channel at low current densities and high temperatures, new flow field designs such as parallel serpentine-baffle flow fields [15], stepped flow fields [16], and fractal foliated flow fields [17] have been studied. However, at high current densities, the accumulation of water in the rib-channel structure leads to the blockage of reactant transport and lower water removal capacity [18,19].

Porous metal foam (PMF), with light weight, high strength, and rigidity, has been considered as an ideal medium for heat transfer or thermal management in new energy fields [20]. Murphy et al. [21] first used two flat porous metals, nickel foam and expanded titanium, as the flow fields embedded in the anode and cathode for a low-cost, lightweight fuel cell stack with a high power density. It was widely proven that a metal flow field was able to improve the performance of a PEMFC by bringing a proper water removal ability that enhanced the uniformity of temperature and humidity in the fluid [22,23,24,25,26,27]. Ahn et al. [28] compared the performance of a PEMFC with copper foam and serpentine flow field bipolar plates under 20% gas humidification conditions. The results showed that the PMF flow field exhibited favorable single-cell performance under pressurized and low-humidity conditions, where the capillary phenomenon could remove excess generated water from the electrode through wicking and humidify under-saturated gas streams through evaporation [29]. Cheng et al. [30] used a PMF flow field in an alkaline anion exchange membrane to analyze the transports and distributions of reactants and water. The porous flow field was proven to be beneficial for membrane hydration, anode water removal, cathode water utilization, and reactant distribution. Bao et al. [31] used a two-phase volume of a fluid model to investigate the gas transport and liquid water dynamics. The results showed that the porous flow field could avoid total blockage of the flow field and reduce the water accumulation.

Therefore, a PMF flow field with a complex foam structure could increase the contact area between the flow field and gas diffusion layer, while improving the distribution of reactant gas and the removal of water. However, there are few studies on the effect of PMF flow fields under flooding conditions, especially by experimental methods. In this work, the fuel cell performance under different water content conditions is experimentally investigated. In particular, the water management of the PMF flow field at the cathode and anode under various flooding situations is examined to understand the dynamic behavior of the fuel cell by various electrochemical characterizations. The mechanism of water transport in the electrode and membrane during the fuel cell operation is therefore discussed, as shown in Figure 1.

## 2. Experimental

### 2.1. Single-Fuel-Cell Test System

The membrane electrode assembly (MEA), consisting of carbon papers (HCP120, Shanghai Hesen Electric Co., Ltd., Shanghai, China) and a catalyst-coated membrane (CCM, Jiangsu GPTFC System Co., Ltd., Kunshan, China) with a Pt loading of 0.3 mg cm^−2^ at the cathode and 0.2 mg cm^−2^ at the anode, was hot-pressed at 135 °C and 0.15 MPa, for 3 min. A simplified single cell, only containing heating elements, end plates with grooves for the flow field, gaskets, and an MEA, was assembled. The Ni foam pieces (Kunshan Lvchuang Electronic Technology Co., Ltd., Kunshan, China), with a thickness of 5 mm, porosity of around 95%, pore size of 450 μm, and surface density of around 1500 g∙m^−2^, were embedded in the end plate for the flow field at the anode and cathode. The contact area between the flow field and MEA was around 5.49 cm^2^, which was recognized as the real surface area [32].

### 2.2. Electrochemical Tests

In the fuel cell test system, as shown in Figure 2, hydrogen and nitrogen from gas cylinders were supplied to the anode through a pressure-regulating valve and a gas flow meter. N_2_ was supplied to the anode first to purge the system with a flow rate of 200 mL min^−1^ for 10 min. Next, hydrogen was pressurized and fed to the anode at a flow rate of 50 mL min^−1^. In the meantime, the air was pressurized by an air compressor and then fed to the cathode at a flow rate of 100 mL min^−1^. Two heating pads on the anode and cathode were used to control the fuel cell operating temperature at 60 °C. The single cell was tested without inlet humidification.

To study the fuel cell performances under different flooding conditions, various amounts of distilled water were sprayed uniformly onto the surface of the nickel foam with a spray gun. As the flow volume of PMF was around 4.29 cm^3^ [32], the amount of water, 0.5, 1.0, 2.0, and 2.5 g, accounted for 16.7%, 23.3%, 46.6%, and 58.3% of the flow volume, respectively. The nickel foam without hydrophobic treatment was tested to behave with high hydrophilicity. Once the water was dripped onto the surface of the tested material, it dispersed too fast to capture the transient image.

An *I*–*V* curve was obtained by adjusting the potential variations, with a voltage limit of 0.25 V. During the constant-potential test at 0.6 V, the initial state was recorded after 2 min of operation and then the data were recorded every 10 min.

Electrochemical impedance spectroscopy (EIS) was employed on an electrochemical analyzer (CHI660E, Shanghai Chenhua Instrument Co., Ltd., Shanghai, China), in the region of 10 kHz to 5 mHz, at 0.7 V with 5 mV amplitude.

## 3. Results and Discussion

### 3.1. Effect of Water Addition

The effect of water amount in the PMF on the electrochemical performance at the initial state is studied at the anode and cathode. As shown in Figure 3, the current density decreases with the amount of water addition, either at the anode or at the cathode. In the low-current-density region (lower than 0.4 A cm^−2^) of the polarization curves in Figure 3a, water addition at the anode has a slight influence on the electrochemical-activation-controlled reaction [33]. A larger difference is depicted at high current densities because of the diffusion resistance. Water stuck in the pores of nickel foam hinders the gas transfer and, therefore, less reactant could be transferred to the surface of the catalytic layer for electrochemical reaction, resulting in poor electrochemical performances [34]. However, the polarization curves at the cathode vary considerably with the water addition in the whole region. The reason is that excessive water product accumulates in the cathode and less reactant is transported for electrochemical reaction, eventually causing the cell to flood quickly [35]. The maximum current decreases by 24.1% and 62.5% at the anode and cathode, respectively, with the water addition of 2.5 g. Comparatively, more serious flooding is depicted at the cathode, which is attributed to a higher stoichiometric ratio of air.

After various water content additions, the electrochemical impedance spectra at 0.7 V are shown in Figure 4. The intersection at the *Z*′ axis is attributed to the ohmic resistance, which increases with the water amount both at the anode and the cathode. While the ohmic resistance consists of the proton transport resistance in the membrane and the electron transport resistance in other components, a higher ohmic value at higher water additions may cause a higher contact resistance instead of higher membrane resistance. As depicted in Figure 3, the water amount only affects the PMF properties at the initial state, while the membrane mostly works under relatively dry conditions because of the absence of inlet humidification. As a high volume of PMF flow field is occupied by the water addition, the path for electron transport decreases, which results in a higher contact resistance between the PMF flow field and the electrode. In Figure 4a, the electronic impedance spectra are all semi-circular when the water addition is from 0.5 g to 2.5 g. The diameter of the loop increases from 22 to 27 mΩ cm^2^. In Figure 4b, semicircle loops are depicted as the water amount is 0.5 and 1.0 g, and the diameter increases with water addition. In contrast, double-arc or even triple-arc lines are observed as the water amount is above 1.0 g, indicating complex internal impedance in the fuel cell reaction [36].

### 3.2. Effect of Constant-Potential Test Time

Under an operating temperature of 60 °C without inlet humidification, the fuel cell performances changes with the constant-potential test time, as various amounts of water are added into the anode and cathode. The polarization curves and impedance spectra every ten minutes are recorded until an obvious current decrease is depicted to avoid deadly damage to membrane materials.

The polarization curves at the initial state are compared with those after the constant-potential testing, with a water addition of 0.5, 1.0, 2.0, and 2.5 g, as presented in Figure 5. In Figure 5a, the current increases in the first 10 min and then begins to decrease. Similar performances are illustrated in Figure 5b–d, while the time required to reach the optimum state is 20, 40, and 50 min, respectively. The time taken increases with the addition amount of water, as well as the maximum current. The maximum values in Figure 5c,d reach around 1.0 A cm^−2^. During the constant-potential test without inlet humidification, the excess water in the PMF is supposed to be removed gradually to resolve the flooding problem and humidify the membrane, and therefore the fuel cell performance is enhanced. However, the current densities decrease with time after the optimum state because of the insufficient humidification of the membrane.

In Figure 6, the effect of the operation time and water amount in the anode has also been studied by EIS. The impedance parameters were obtained by fitting with the Randles circuit [37,38], consisting of ohmic resistance (*R*_Ω_), charge transfer resistance (*R*_ct_), and constant capacitive component (*C*_d_). The values of *R*_Ω_ and *R*_ct_ are given in Table 1.

As depicted in Figure 4, no obvious diffusion resistance is shown in Figure 6a–d. In Figure 6a, as 0.5 g of water is added to the PMF, no significant change with time is shown in the impedance spectra. The value of *R*_Ω_ is equal to 34 mΩ cm^2^. In Figure 6b–d, an obvious decrease in polarization impedance, including *R*_Ω_ and *R*_ct_, is observed with the single-cell operating time. The maximum values, 39 mΩ cm^2^ (*R*_Ω_) and 30 mΩ cm^2^ (*R*_ct_), are obtained as 2.5 g of water is added at the initial state. The minimum values, 34 mΩ cm^2^ (*R*_Ω_) and 17 mΩ cm^2^ (*R*_ct_), are obtained at the optimum state as 2.0 and 2.5 g of water are added after 40 and 50 min of testing, respectively. A lower *R*_ct_ reflects higher catalyst activation and better hydration of the membrane in the catalyst layer [39]. As the test time further increases, the resistance values increase because of the drying membrane.

The effect of the constant-potential time and the water addition in the cathode has also been discussed and illustrated in Figure 7. As displayed in the anode, the maximum current also increases with time as various addition amounts of water are added to the cathode flow field. The time required to reach the optimum state increases with the addition of water, equal to 20, 40, 50, and 50 min. Compared with the current density in Figure 4, a lower current density value is obtained, as shown in Figure 7, while the maximum value is around 0.8 A cm^−2^ with 2 g and 2.5 g of water in the cathode flow field tested after 40 and 50 min, respectively.

In Figure 8, Nyquist plots of the single cells with PMF after various amounts of water addition under the constant-potential test are presented. No obvious flood performance is observed in Figure 8a with 0.5 g of water added to the cathodic flow field. The impedance loop shrinks slightly with time, indicating a decreasing charge resistance. As depicted in Figure 4b, at the initial state, serious diffusion is seen after more than 1.0 g of water addition, as shown in Figure 8b–d. This phenomenon is also alleviated gradually with the operation of the constant-potential test. The diffusion loops disappear after 20, 40, and 40 min, after 1.0, 2.0, and 2.5 g of water addition, respectively.

The diffusion resistance is considered in the equivalent circuit [40], consisting of ohmic resistance (*R*_Ω_), charge transfer resistance (*R*_ct_), and a constant capacitive component (*C*_d_); the Warburg impedance (*Z*_w_) is determined by the resistance of mass transfer; and the constant-phase element (*Q*) represents the electrical double-layer capacitance. The values of *R*_Ω_ and *R*_ct_ are listed in Table 2. The lowest values of *R*_Ω_ and *R*_ct_ are obtained at the optimum state, 35 and 21 mΩ cm^2^, respectively, while no diffusion loop is found.

The variations in maximum current density, *R*_Ω_, and *R*_ct_ with the addition time of water are illustrated in Figure 9. The fuel cell performances under anode and cathode flooding conditions are compared in Figure 9a,b, respectively. At the initial state, the large current drop is due to a high accumulation of water in the flow field and electrode, which reduces the ability to transport reactant gases into the catalyst layers. Greater abilities of anti-flood and mass transfer and higher current densities are found as water is added to the anode. In Figure 9c, the value of *R*_Ω_ varies in the range of 34–37 mΩ cm^2^, which is slightly lower than that in Figure 9d, between 34 and 42 mΩ cm^2^. The value of *R*_ct_ in Figure 9e is significantly lower than that in Figure 9f under the same water addition amounts. As the cell generates water at the cathode, the oxygen supply would be impeded by liquid water [41].

## 4. Conclusions

The water management capacity is of great significance for the performance of the single cell. Obvious current decay was observed after water addition to both the anode and cathode. The rate reached 24.1% and 62.5%, as 2.5 g of water addition was added to the anode and cathode PMF flow field, respectively. Greater abilities of anti-flood and mass transfer were found as the water was added to the anode, where no diffusion loop was depicted in the impedance plots. The effect of operation time and water amount in either anode or cathode was further studied by a polarization test and EIS. The phenomena of water flooding were alleviated by water transferring to humidify the membrane and purging the residual gas during the constant-potential test. A relatively high current of 1.0 A cm^−2^ was obtained at the optimum state as the same amount of water was added into the anodic flow field, which may be due to a lower *R*_ct_, around 17 Ω cm^2^. The PMF flow field revealed good water capacity despite the high-water content in the flow field, because of the capillary phenomenon in the pore structure of the PMF. The porous metal pores stored a certain amount of water, which entered the electrode and membrane for wetting treatment, to achieve an internal “self-humidification” function. The proton conductivity of the proton exchange membrane was enhanced, thus improving the PEMFC performance.

## Figures and Tables

**Figure 1 micromachines-14-01224-f001:**
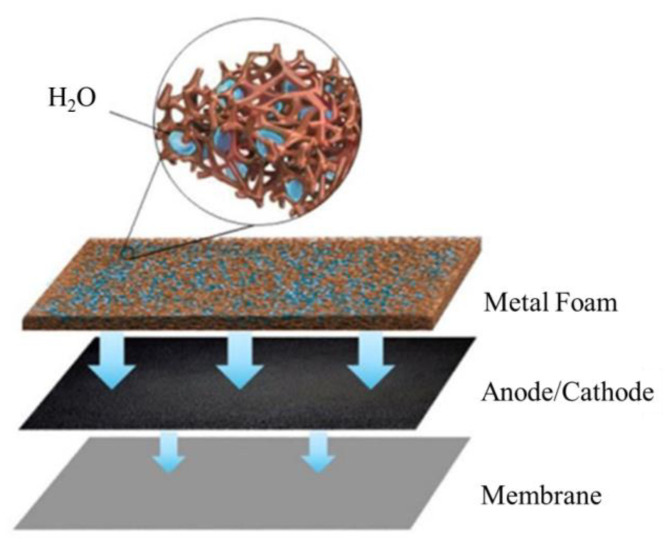
Schematic diagram of porous flow field under humidification conditions.

**Figure 2 micromachines-14-01224-f002:**
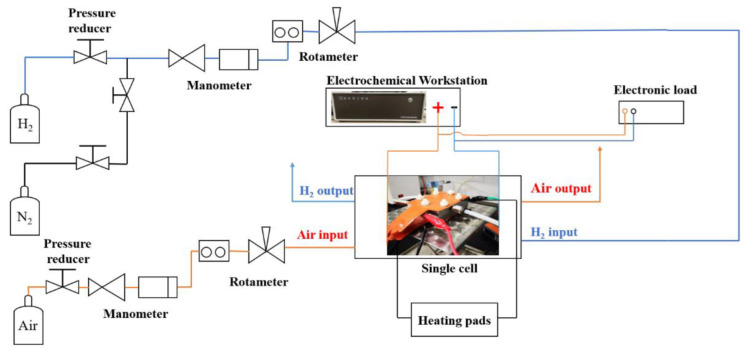
Scheme of the single fuel cell test system.

**Figure 3 micromachines-14-01224-f003:**
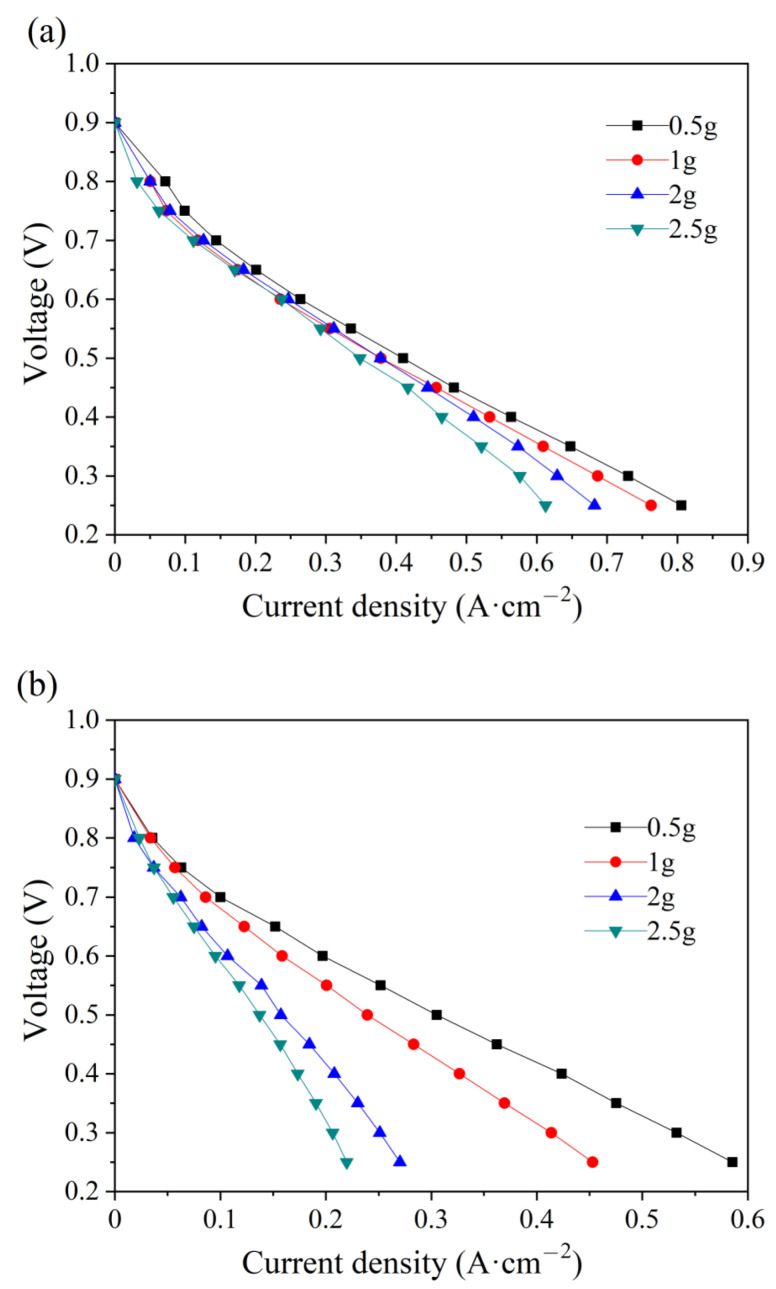
Polarization curves of single cells with different water additions into the anode (**a**) and cathode (**b**) flow field at the initial state.

**Figure 4 micromachines-14-01224-f004:**
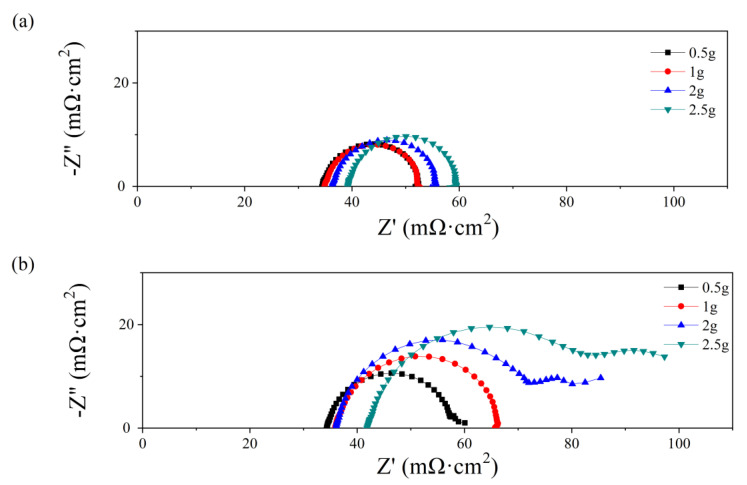
Nyquist plots of single cells with different water additions into the anode (**a**) and cathode (**b**) flow field at initial state.

**Figure 5 micromachines-14-01224-f005:**
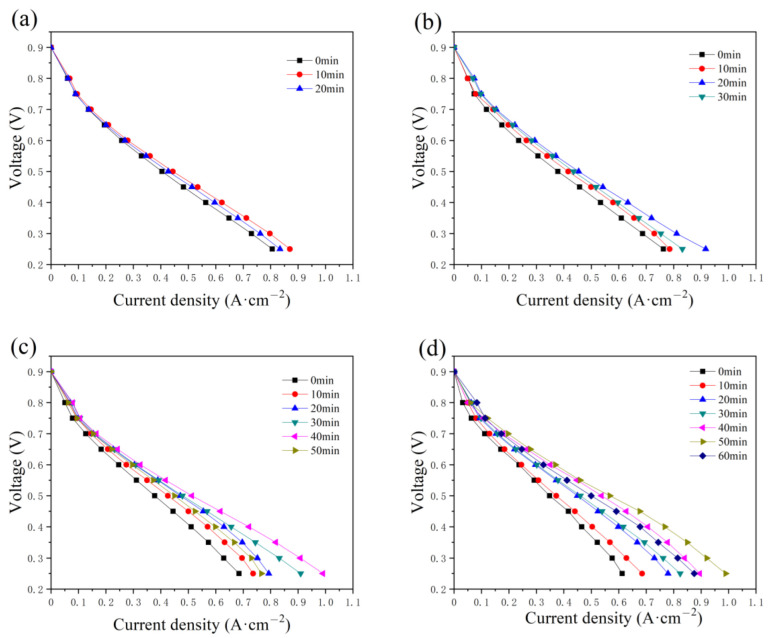
The variation in the polarization curves with the constant-potential time as (**a**) 0.5 g, (**b**) 1.0 g, (**c**) 2.0 g, and (**d**) 2.5 g of water are added into the anode flow field.

**Figure 6 micromachines-14-01224-f006:**
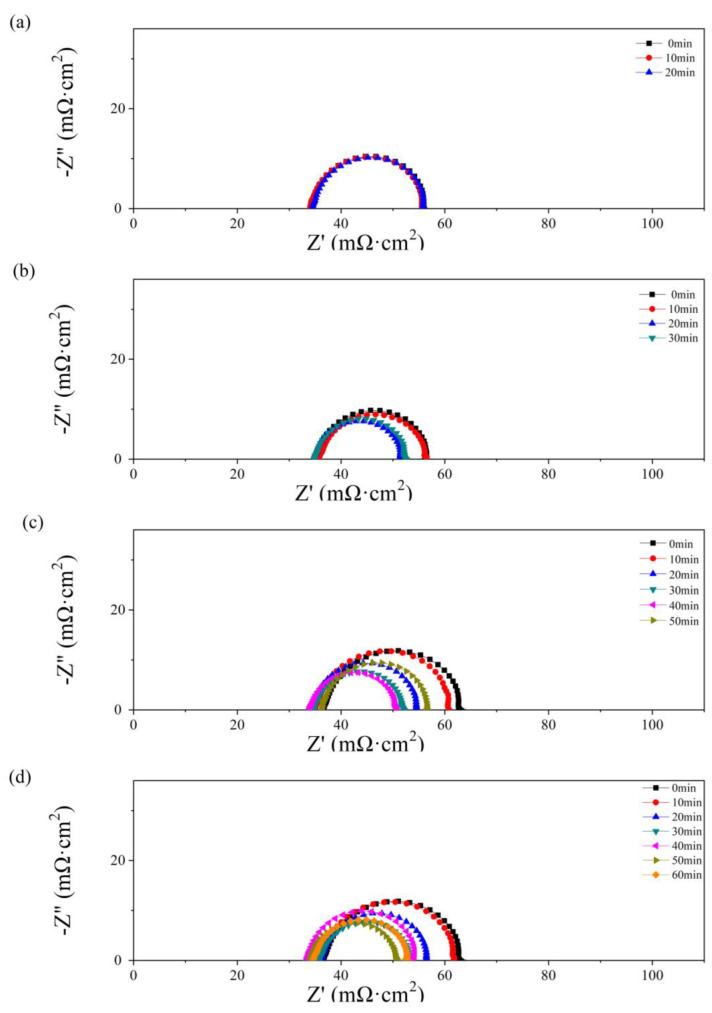
Nyquist plots of the single cells with 0.5 g (**a**), 1 g (**b**), 2 g (**c**), and 2.5 g (**d**) of water added to the anode flow field after different operating times.

**Figure 7 micromachines-14-01224-f007:**
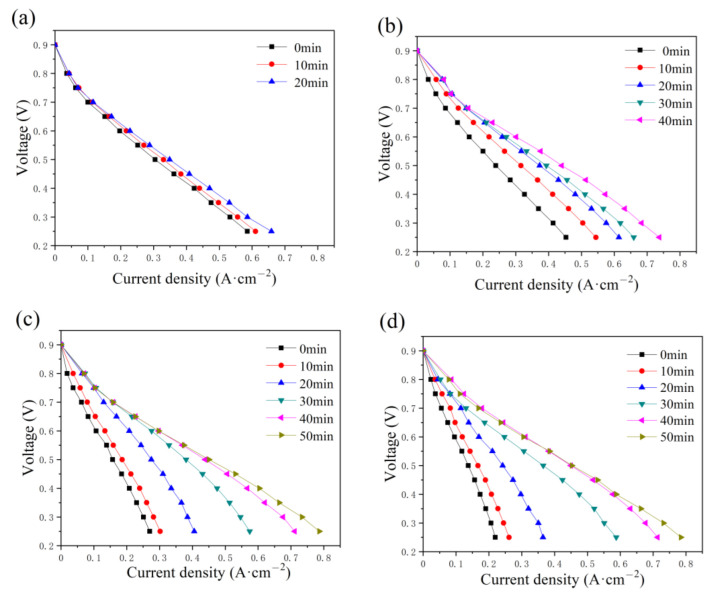
Polarization curve changes with constant-potential time as (**a**) 0.5 g, (**b**) 1.0 g, (**c**) 2.0 g, and (**d**) 2.5 g of water are added into cathode flow field.

**Figure 8 micromachines-14-01224-f008:**
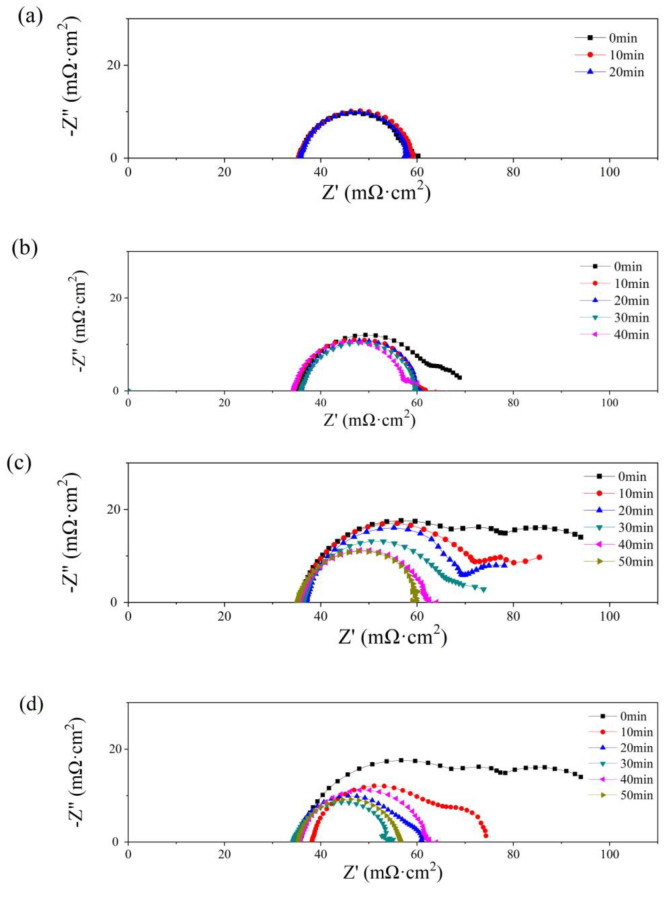
Nyquist plots of single cells with 0.5 g (**a**), 1 g (**b**), 2 g (**c**), and 2.5 g (**d**) of water added into the cathode flow field after different operating times.

**Figure 9 micromachines-14-01224-f009:**
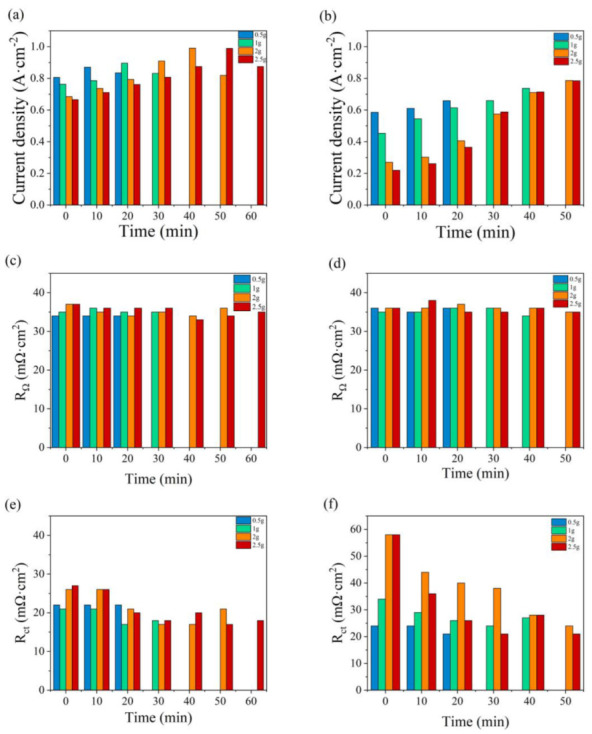
Variations in the maximum current density, *R*_Ω_, and *R*_ct_ with the water addition time at the anode (**a**,**c**,**e**) and at the cathode (**b**,**d**,**f**).

**Table 1 micromachines-14-01224-t001:** The values of *R*_Ω_ and *R*_ct_ (Ω cm^2^) of the single cells with 0.5, 1, 2, and 2.5 g of water added to the anode flow field after different operating times.

Water Amount/g	T/min	*R*_Ω_/mΩ cm^2^	*R*_ct_/mΩ cm^2^
0.5 g	0	34	18
	10	34	22
	20	34	22
1 g	0	35	18
	10	36	21
	20	35	17
	30	35	18
2 g	0	36	11
	10	35	26
	20	34	21
	30	35	17
	40	34	17
	50	36	21
2.5 g	0	39	30
	10	36	26
	20	36	20
	30	36	18
	40	33	21
	50	34	17
	60	35	18

**Table 2 micromachines-14-01224-t002:** The values of *R*_Ω_ and *R*_ct_ (Ω cm^2^) of the single cells with 0.5, 1, 2, and 2.5 g of water added to the cathode flow field under different operating times.

Water Amount/g	T_cp_/min	R_Ω_/mΩ cm^2^	R_ct_/mΩ cm^2^
0.5 g	0	34	26
	10	35	24
	20	36	21
1 g	0	36	30
	10	35	29
	20	36	26
	30	36	24
	40	34	27
2 g	0	36	44
	10	36	44
	20	37	40
	30	36	38
	40	36	28
	50	35	24
2.5 g	0	42	55
	10	38	36
	20	35	26
	30	34	21
	40	36	28
	50	35	21

## Data Availability

Data will be available upon request from the corresponding authors.

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
