# Peer review of "Water Management Capacity of Metal Foam Flow Field for PEMFC under Flooding Situation"

_micromachines, 2023, doi:10.3390/mi14061224_

Round 1

Reviewer 1 Report

The manuscript describes the experimental study of the PEMFC performance with metal foam flow field, in particular the performance under the flooding conditions. Polarization curve and EIS are used to evaluate the performance.

However, the description of the experimental setup is not clearly presented and the obtained results are not well analyzed nor explained. Therefore, the reviewer believes the manuscript is suitable for the journal publication.

More specifically, some (but not all) major issues of the manuscript are listed below.

1) In Section 2.1 it is written that the cell consists of "only containing heating elements, end plates, gaskets, and MEA".  Then, where does the metal foam come into play?

2) In Section 2.2 it is written that the various amounts of distilled water were sprayed uniformly onto the surface of the nickel foam. Does this mean the metal foam is exposed to the atmosphere, not enclosed? If enclosed, how can the water be sprayed?

3) Again about the water spray (Section 2.2). The amount of water (0.5g, 1.0g, etc.) is just one time, or is it a water injection rate per unit time?

4) Figure 3. How is the polarization curve at 0g water spray? It must be important for the comparison.

5) Figure 4. In the Nyquist plot, as the authors well understand, sum of the proton transport resistance in the membrane and the electron transport resistance in other components (which is relatively small) can be found at the high frequency side intersection with the x-axis. Then, why the proton transport resistance increases as the amount of water spray increases? It seems to the reviewer the proton transport resistance decreases with the water spray since the membrane water content increases.

6) Figure 5. In all 4 cases the performance first increases and then decreases. Why is it? Is the performance drop at the end recoverable?

7) Now in Figure 7, the performance keeps increasing with time. Why is it? What happens if the cell is operated longer time?

There are not many spelling or obvious grammatical errors in the text. In some locations, however, the selection of the terms is not appropriate (for example, "dispersion" in line 11, "flood resistance" in line 16). Connections between sentences and between paragraphs can be also improved.

Reviewer 2 Report

Chen et al. constructed PEMFCs using porous metal foam (Ni foam) as the flow field and systematically investigated the influence of different water content for current densities and resistances in anode and cathode. To some content, this is a scientific and systematic paper focusing the crucial issue—flooding phenomena in PEMFCs. It might be accepted for publication in Micromachines after following revisions.

1. The authors claim that the Ni foam as flow field will be very beneficial to the removal of water and thus PEMFCs performance. However, the comparison for current densities between the Ni foam and conventional flow field is lacking and need be added.

2. Ni foams (5 mm) was used as flow field in the paper. The reasons about selecting the thickness or the comparisons between varied thickness is encouraged to complement.

3. The write of “A/cm2” or “A∙cm-2” should be unified in the paper.

Reviewer 3 Report

Micromachines

 Manuscript: micromachines-2437024

 Title: Water management capacity of metal foam flow field for PEMFC under flooding situation

 Authors: Lingjiang Chen, Zichen Wang, Chuanfu Sun, Hui Zhu, Yuzhen Xia, Guilin Hu and Baizeng Fang

 Summary

The manuscript presents studies of PEMFCs using porous metal foam (PMF) as flow field. The authors evaluated effects of PMF flooding on fuel cell performance. It was determined that cathode PMF flooding caused performance drop compared to anode flooding. The authors noted that water in PMF served as a source of internal humidification when the fuel cell was operated with dry inlet gas supply.

My recommendation is Accept after minor revision.

 Questions and comments.

1.     Experimental. Experimental section does not clearly describe the performed work.

1.1. Was PMF used at anode and cathode? What are parameters of PMFs, like pore size, pore volume etc?

1.2. What is the compression ratio of the MEA?

1.3. Operating conditions. I understand that the IV curves were measured at fixed flow rates of H2 and Air of 50 and 100 ml/min. I believe that for 5.49 cm2 MEA this flow rate is not enough to ensure stoichiometry 2 at current density of 1 A/cm2. So, at high current operation you will be at lower stoichiometry than at low current and this will affect the performance.

1.4. What were inlet gas humidification and back pressure?

1.5. Page 3, lines 109-111. As the flow volume of PMF was around 4.29 cm3, the amount of water, 0.5, 1.0, 2.0, and 2.5 g, accounted for 16.7%, 23.3%, 46.6%, and 58.3% of the flow volume, respectively.” It is not clear what was meant by this sentence. I assume that the PMF was somehow impregnated by different volumes of water to simulate certain flooding conditions from 16.7 to 58.3% of the PMF volume. Please clarify.

1.6. I am wondering how it was possible to keep this amount of the PMF flooded and not change during IV measurements especially if you supplied dry inlet gas? I believe that gas flow through the PMF will affect the amount of water stored and the obtained data is impossible to interpret since your experiments were not performed at steady state conditions.

1.7. Page 4, lines 119-121. “Electrochemical impedance spectroscopy (EIS) was employed on an electrochemical analyzer (CHI660E, Shanghai Chenhua Instrument Co., Ltd, China), in the region of 10 kHz to 5 MHz, at 0.7 V with 5 mV amplitude.” What was the frequency range? It is typically from 10 kHz to 0.1 Hz.

1.8. The EIS was performed at voltage control mode (0.7 V). However, I would strongly recommend working with EIS under galvanostatic control. If you measure EIS at the same current you can correctly compare the data since you are at the same reaction rate. While under potentiostatic control you are getting different current at 0.7 V meaning different reaction rate and different dominating processes (like kinetic vs mass transport).

1.9. Why do not you show a reference results for the PMF without any water content?

2.     Results and discussion

The presented results are interesting, but the authors need to state clearly what was the purpose of the experiments. In my opinion, it is a dynamic behavior of the fuel cells with different level of flooded PMFCs. It would be very beneficial if the purpose of the paper would be more specific. For example, “In this work, the fuel cell performance of PEMFC under different water content conditions is experimentally investigated. In particular, the water management of the metal foam field at the cathode and anode under flooding situations is examined to understand dynamic behavior of fuel cell performance operated with dry gas.”

It looks like the anode or cathode PMF was initially flooded and when dry gases were fed and IV or constant voltage hold (CVH) @ 0.6 V were executed. During operating, the water content in the PFM reduced, water moved to MEA and humidified the membrane resulting in a decrease in HFR (ROhm). At the same time, the removal of water from the PFM opened flow paths for air or H2, leading to improvement in mass transport which is observed by EIS after 60 min of CVH. I think it would be very interesting to look at plots/figures illustrating current response vs time at CVH for the cells with different flooding of the PMF.

Round 2

Reviewer 1 Report

Thank you for providing detailed responses to my review comments.

I think the manuscript is ready to publish with some minor corrections. For example,

1) In the Abstract and in the Conclusions, I believe the charge transfer resistance is 17 mohm cm2, not 17 ohm cm2.

2) In line 181, the maximum current density should be 1.0 A/cm2, not 1.0 mA/cm2.

3) Similarly, in line 214. 0.8 mA/cm2 should be 0.8 A/cm2.

Nothing to be added.

Reviewer 2 Report

Authors have addressed most of my concerns and the paper can be accepted for publication.